# Teaching Spatial Planning Using Elements of Design Thinking as an Example of Heuristic in Urban Planning

**Rafał Blazy \*** and **Mariusz Łysień**

Faculty of Architecture, Cracow University of Technology, Warszawska 24, 31-155 Cracow, Poland;
mlysien@pk.edu.pl
\*  Correspondence: rblazy@pk.edu.pl

**Abstract:** In modern urban analysis, line–nodal connections are often used as a model, diagram or matrix for describing space and its accompanying phenomena. In practice, the search for relationships in space is often nothing but the designation of lines reflecting their physical and nonphysical association by their geometry. Recording the spatial issues of a city constituting a very complex structure on the surface of paper, or on a computer monitor, is an extremely difficult conceptual task. At the heart of Design Thinking is a deep understanding of the relationship between different elements. It can therefore be concluded that heuristics are the basis for understanding the Design Thinking method. An attempt was made to check whether the Design Thinking method can restore the optimal balance between modern tools used to develop course projects, and the need to search for the best solution constituting the idea of the project. The article presents the results of two-year research on the use of the Design Thinking method in academic teaching of subjects related to the subject of spatial planning. Thanks to the use of the Design Thinking method in teaching spatial planning, it is often possible to discover design possibilities and proposals that surprise students.

**Keywords:** Design Thinking; urban planning; teaching; city; spatial planning

## 1. Introduction

Heuristics (from Greek word εὑρίσκω—heuriskō—which means "find") is the ability to detect new facts and relationships between them, often using previously to made hypotheses. In spatial planning, this is usually done by graphical analysis, which is an instrument for mapping and codifying three-dimensional reality. These analyses are to help in getting to know and presenting complex issues such as the city, the region or even the whole country. In addition to the abovementioned facts (in philosophy phenomenon), space is also sought for relationships and associations that exist in a given area. Attempts to graphically reproduce them are most often made by symbolic connection of individual elements of the spatial and functional structure by and with the lines and axis. This method is looking for spatial relations, and, at the same time, it illustrates the way of thinking about the problem by its authors, reminiscent of the method called "mind-mapping".

At present, cities are developing in a highly dynamic manner. This is the result of numerous factors, which include social transformations with a particular emphasis on changes in the mode of working and the modernization of production processes in rural areas. It is necessary to understand what urbanization is and on what levels it is taking place [1]. The pursuit of better financial conditions in urbanized areas is linked with a boarder offering and supply of employment. This leads to the rapid development of urbanized areas which often do not possess proper planning documents. Similar tendencies can be observed all around the world [2]. The considerable demand for such documents places strict requirements before spatial planning [3–5]. The proper training of graduates in this field is necessary to ensure the proper quality of planning documents that they will prepare.

## 2. Research Background

In practice, the search for relationships in space is often nothing but the designation of lines reflecting their physical and nonphysical association by their geometry. In this process, the geometry and form that best reflect the excavated reality are searched for. The method, which simply connected points of the system that possessed the similar characteristics or some specific feature or relation to each other, belongs to the first group of analyses, from the point of view of their complexity and the accompanying calculating operations. Basically, it examines spatial correlations that determine the relationship—links between individual elements of the system. Some of the analytic functions are replaced by graphic operations consisting in showing, for example, communication between places in a analyzed space. These studies serve to characterize the spatial distribution and functional-spatial relationships in order to visual form [6]. The visualization of these interrelations and relation is the primary goal of the notation. Indirectly through this type of exploration, the topological properties selected for the study of places are determined.

From the graphical point of view (Figure 1), such an analysis belongs to vector analysis, determining also the distance between the points occurring and their strength and direction, and the type of their interaction.

In this method, objects are uniquely associated with their attributes that are used to classify them. The entire process consists of the following two parts:

1. Designation of objects, groups, regions or areas with their possible classification, due to their attributes.
2. Creating a grid—line junctions and links between individual elements.

In an attempt to more accurately record, it also includes information about the types of relationships and how they connect with other objects. In the drawing, there is a graphical integration of data, showing points and important modal places—the culmination in a given area. In this way, the validity and hierarchy of given places' centers is indirectly established [7].

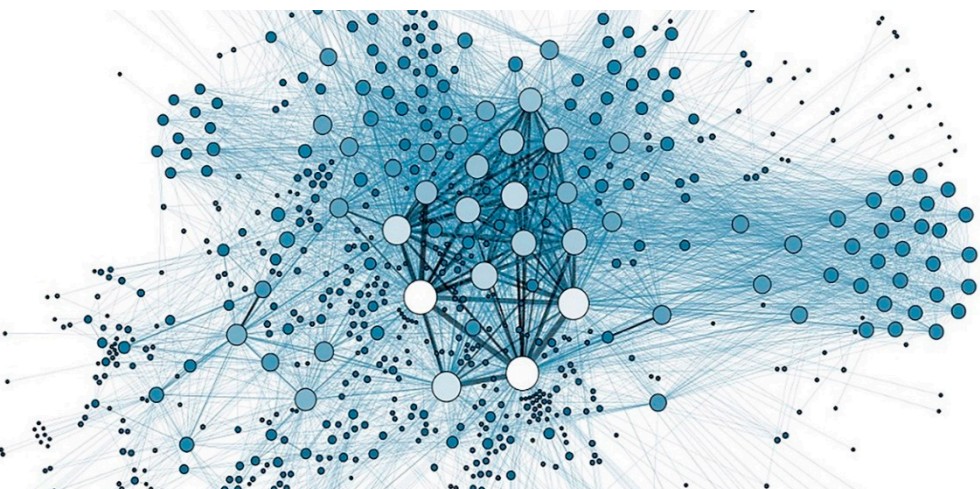

**Figure 1.** An example of a network analysis, "Social, Business and Political Research". Source: https://www.acspri.org.au/sites/127.0.0.1/files/Social_Network_Analysis_Visualization-1000x458.png (accessed on 14 July 2020).

Considering that our thinking is based on mental structures and imaginative structures, it is much easier for us to think about a given problem with its visual representation. In this way, all kinds of diagrams and abstract images of spatial relations are treated as internal (personal) and external (interpersonal) communication tools, facilitating communication and improving their readability and provide spatial detailing analyses. The very basic concept of planning is undoubtedly connected with the sketching aspect of spatial organization, so it is inseparably connected with the image and some geometrical drawing representation.

The thinking process can be said that in its essence it is very analogous to the discussed spatial analyses. It constitutes a specific structure consisting of building components and the way (rules) of combining them [8]. Considering the similarities of these two processes, we can put forward the thesis that the analysis of spatial conclusions is the extraction and strengthening of our way of thinking about the problem, and thus some kind of urbanist's mind mapping thinking.

Decision psychologists distinguish the process of solving problems as algorithms such as the following:

- Decision tree—representing the task in the form of a tree diagram. A decision tree requires imagining the initial state, and then possible steps, and assessing the consequences of each step along with making the choices and comparing with the desired state. The root of the tree is the starting situation, further branches symbolize variant choices, and they then branch out, with only one of these branches reaching the desired goal and optimal solution.
- Decomposing problem—breaking down the problem into more detailed problems, which then breaks down into even more detailed and retail aspects, etc. [8].

In the context of spatial analysis of cities, the form of manifesto was published in the 1966 article "A city is not a tree" by contemporary theoretical architecture and urbanism researcher—Christopher Alexander. The author expressed his opposition to the approach that brought the cities to simple, hierarchical, easy-to-analyze and interpret structures. He therefore criticized all other utopias that aimed at simplifying and reducing the holistic issue of the city [9]. Alternatively, Christopher Alexander proposed a method for studying and shaping cities using the complex network model of half-nets. He claimed that the lack of basic structural complexity, characteristic of tree diagrams (Figure 2), paralyzes and limits our imagination about the cities [9]. He explained that the generally accepted simplified structural models lack the ability to describe the entire socio-spatial complexity of cities.

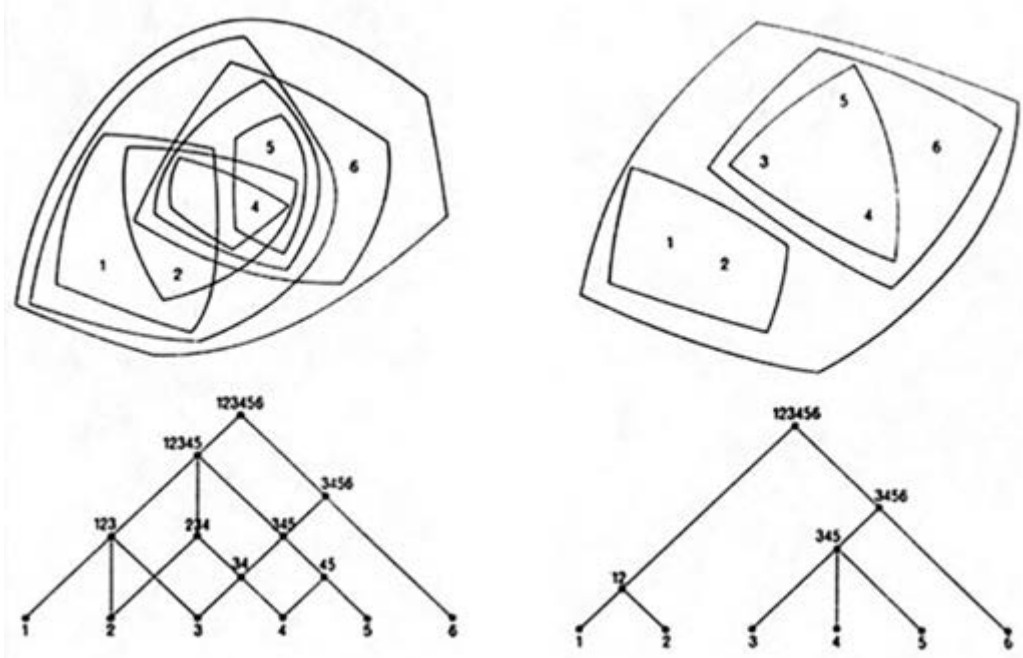

**Figure 2.** Illustration of Christopher Alexander, "City is not a tree": natural cities (upper row) and artificial cities (bottom row). Source: https://www.europan-europe.eu/media/default/0001/13/39b6c310e1822d23752a7eec2888d1e33e2e8421.png (accessed on 8 October 2020).

Building on a basic of one-dimensional simple network of structural connections in which all junctions are serial, especially in planning and design practice, is the source of

many errors and distortions. Christopher Alexander was convinced of the following: "In every city there are a thousand, even a million times more active and complicated systems, whose physical products do not fit in 'trees structures'. In the worst case, the units that fall into these structures lack references to the existing reality, and the real systems whose presence is the basis of urban life are devoid of physical context". This theoretician believed that the adoption of the network model is not tantamount to stopping the search for order, but rather is the perception of a more complex order, in his book *The Nature of Order*, he resisted, among others on the theory of Stephen Wolfram presented in the book *A New Type of Learning* [9].

In modern urban analysis, line–nodal connections are often used as a model, diagram or matrix for describing space and its accompanying phenomena. In order to organize some data, systems sometimes seem to be more or less geometrical, which simplify some of the processes occurring in an analyzed area. The synthesis of this record seems to be at the same time its strength—as simplicity and as weakness—as an excessive simplification.

The mapping of the simplest relations, bringing the problem to the graphic symbol— on the one hand, it is a very helpful solution, but on the other hand, it threatens with reductionism of the problem complexity. Sometimes the network of connections and interactions in urban concepts creates an autonomous form of scheme creation—independent of the environment and surroundings (Figure 3). Although it facilitates large-scale activities by filling the plan of a given space with abstract lines, it should be remembered that it uses only standard linear elements to characterize the complex and variant system of the city. However, from the position of a walking man around the city, it is completely illegible and invisible.

Linear connections from point to point (from place to place) are something really artificial in urban space. They are very rare in natural forms. For the most part, in the natural environment, they are distorted—bent, wavy or interrupted—and they are more humanistic. It should be remembered that human life on earth is so unique and extremely vital that it is also normally very difficult to grasp in silhouette or pattern, especially when it comes to the issue of space.

Design practice proves that the expert ability to predict facts, circumstances and events is the basis for making design strategic decisions. In turn, the ability to analyze combinations of many factors in the same time allows us to choose the most optimal solution among many variables (from the point of view of circumstances and condition of them). The greater number of variables in the design process to be analyzed, the greater the probability that the chosen solution is not burdened with any hidden error, which, in a particular case, may decide on the sense of a given solution. Not considering even one of the important factors may prevent, suspend or question the validity of the investment.

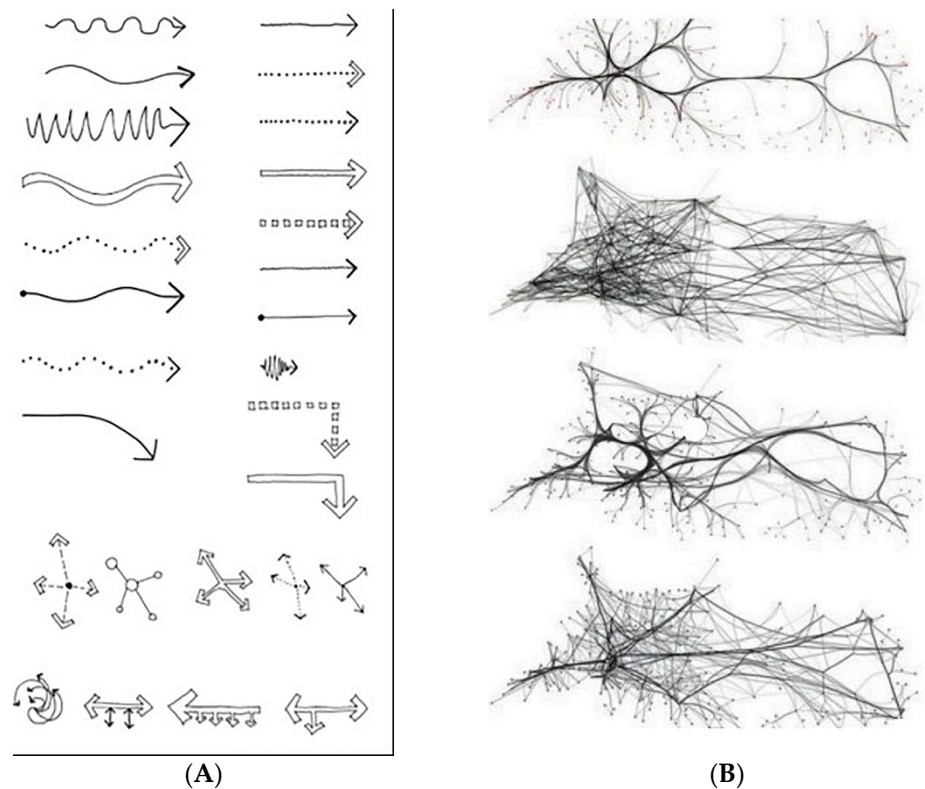

(A)                                                                          (B)

**Figure 3.** (**A**) Modification of the analysis with formally important points—nodes and also line connections for strength and type of interaction. (**B**) Modification of the grid system with soft elements that adapt the grid to the realities of connections and complexity. Source for (**A**)—https://i2.wp.com/www.firstinarchitecture.co.uk/wp-content/uploads/2018/09/arrows. jpg?resize=768%2C779&ssl=1, (accessed on 3 June 2020) (**B**)—http://citeseerx.ist.psu.edu/viewdoc/ download?doi=10.1.1.212.7989&rep=rep1&type=pdf (accessed on 3 June 2020).

For optimal spatial model solutions in the searching, especially in the spatial planning area, a "strategic–structural" approach is usually recommended. In this approach, strategies for influencing changes in urban space should be considered more in structural terms. As for the methodology, it is a combination of three indicated modus operandi: (1) the hermeneutic method, which is applied in the humanities researches and enriched by (2) methodologies recommended in interdisciplinary planning theories; and (3) "structural" approaches to modeling solutions, appropriate technical sciences to which the discipline is assigned, namely architecture and urban planning [10]. Contradictions and ambiguities— discontinuities in the structure of grid and meshes, as well as real connections—often affect the final form of the solution. According to Robert Venturi and his "Complexity and Contradiction in architecture", architectural dissonances are often the source of "picturesque", as also the various meanings. The contradiction, complexity and inconsistency of architecture and the city is the field of search for an urban planner who should extract functions, objects and relations that restore a homogeneous concept of reality from this fuzzy set of data [11].

The research was based on empirical experiments. They are the result of several years of work on implementing Design Thinking (Design Thinking) in the process of teaching subjects related to spatial planning. During them, various work patterns were confronted. The developed methods are proprietary and do not duplicate the diagrams presented in other publications on Design Thinking.

Why Design Thinking? The process of the shaping of a city's structure should comprehensively lead to an improvement of the quality of public space and its associated elements. The course of this process can be generally compared to that of designing buildings. Experience from teaching classes on spatial planning leads to the observation that this subject

matter is difficult for students. This is largely the effect of the curriculum being dominated by design-related modules that feature the design of singular objects rather than entire structures. Planning modules, which often make up a small part of the teaching process, are in opposition to this. It is these types of assignments that define the appearance and functioning of cities around the world. In Poland, the situation is complicated by legal regulations that define the scope and form of preparing acts of local law and that are not without their imperfections [12]. This further highlights the need to properly understand the subject matter and link it with creative and effective shaping of space. Design Thinking follows this convention.

Design Thinking can be briefly described as an approach to creating new products and services based on a deep understanding of user problems and needs (Figure 4) [13]. In this theory, one can define any process associated with universal design or design in general in a similar manner. John S. Gero wrote in one of his works that design exists because the world around us does not appear suitable for us [14]. He discusses various matters associated with the evolution of broadly understood design [15]. Just as in the case of designing a work of architecture or an object of everyday use [16], the process of urban design features a need to gain an in-depth understanding of the needs of the city and the expectations of different social groups. Kees Dorst described Design Thinking as a new and exciting paradigm associated with solving problems in many fields of science [17]. Design Thinking places considerable emphasis on stimulating creativity, making maximum use of existing determinants while often verifying the current effects of the concept under development. The beginnings of formulating the Design Thinking method took place in the 1980s. Among the publications from this period are the works of Peter Rowe [18] or Nigel Cross [19]. Nigel Cross is constantly trying to expand his research on Design Thinking-based design methods [20]. In the literature, Design Thinking is defined as an attempt to innovate, understood as facing challenges that result from creating completely new, better solutions. Christopher Meinel and Larry Leifer highlighted in [21] that the need to innovate is nothing new, but pointed to a considerable increase in the pace at which innovations should be applied in areas of everyday life. They observed that the motto "faster and faster" is becoming a mantra that manifests itself in the functioning of every one of us. This dependence also carries over to design solutions, which are not tested enough prior to being handed over for construction. The authors in question have published several texts associated with the subject of this paper [22–24]. New challenges also pertain to design in compliance with sustainable development, which can allow us to limit adverse climate change. Nature Driven Urbanism, edited by Rob Roggema, is a particularly valuable publication on this subject [25]. It points to the key problems that urbanists face in 2020. The diagnosis is followed by proposals of solutions intended to improve the climate situation around the world and in its major urban centers. Research results [26] confirm that the effectiveness of patterns of use of various patterns and media are dependent on the teaching approach. The improvement of results due to teamwork is also confirmed by other studies [27]. This proves the synergy effect achieved through the involvement of the whole group in solving the task, instead of joining individual parts. This relationship is moreover confirmed by numerous scientific studies on teamwork [28]. Interdisciplinarity and the possibility of using similar patterns in various disciplines have been widely described in many scientific publications [29,30]. This highlights the potential for group action.

The Design Thinking method is supplemented by creating "personas" that reflect unified patterns for specific groups of users. These assumptions arise from the genesis of the method itself, which is the design of everyday items that must meet the needs of the broadest possible group of people which includes different types of consumers. It is recommended that the design team be interdisciplinary, so as to increase its effectiveness.

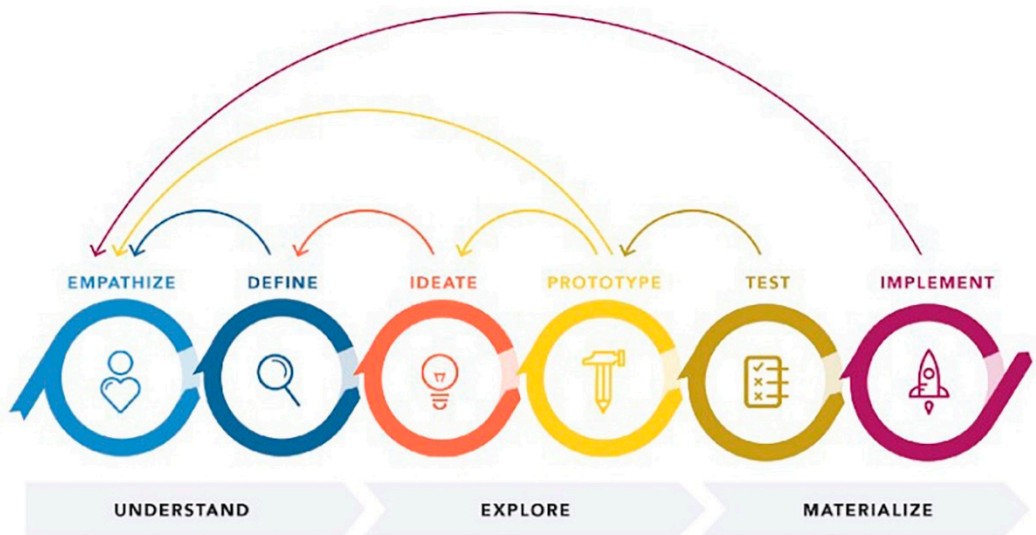

**Figure 4.** General scheme of the functioning of Design Thinking (https://slowapisanego.pl) (accessed on 22 June 2020).

Apart from the initial stage during which needs and assumptions are defined, the method of formulating the conceptual design proposal is also important. We have become accustomed to constant assessment. Design Thinking promotes brainstorming, during which a team generates the greatest possible number of ideas without judging any of them. This results in the appearance of ideas that can sometimes appear to be quite abstractive. As a result, this allows for the production of much more complex designs that often address a diverse array of needs.

The multiple prototyping stage is a critical element of Design Thinking. This means that even a failed result allows the collection of information about elements that need to be improved. In the case of developing a single conceptual design proposal, the elements that are its weak points can be easily overlooked and we can become aware of their existence during the final phase, which can be compared to handing over the design to the client. In the case of multi-stage verification and improvement of conceptual proposals, it is possible to perform a much more precise assessment of the design and whether it meets the assumptions of the assignment. The psychological aspect is an added value here, as the client can feel that the designer is making all efforts necessary to ensure that the design best fits their needs. By frequent contact and the possibility of assessing the proposal by the client, we can build trust between them and the designer. This manner of teaching shapes habits that are not only desirable in professional design but are also largely associated with professions of public trust [31], which often include the architect.

Similar observations were described in the paper entitled "The relation between academic achievement and the spontaneous use of design-thinking strategies" [32]. Its authors highlighted that a considerable portion of design modules is taught in a manner that leads to obvious solutions that do not lead to the development of creative thinking. The end result of such curricula is that students prepare scenarios that insufficiently simulate real-world conditions. The ability to use constructive criticism is also important in the entire teaching process. It is therefore essential to present the proper arguments as to the necessity of introducing changes. Observations noted during teaching spatial-planning modules at the Cracow University of Technology largely correspond to other studies [33] conducted by psychology specialists. Bilateral trust in the will to achieve good results leads to a greater effectiveness of action.

Accounting for the previously mentioned difficulties with translating the scheme of designing a single structure to the scale of an entire urban complex, we will be presented with a situation in which many design proposals appear to be obvious as they are not rooted in a precise analysis of the subject. This scheme is compounded by another factor in the form of the tools the students use. A departure from traditional media towards

computer aided design leads students to give more weight to operating the tool than to pursue new creative and functional solutions. It also negatively affects the linkages between each elements of the city's structure or the analysis of compositional elements. As a part of the POWER project, the Cracow University of Technology organized a cycle of training courses for a group of its employees under the name "Architectural Device Hub < 35", and it featured a module on Design Thinking. It appeared to be an interesting alternative whose application in the process of teaching spatial planning could carry over to improving the quality of assignment projects.

One can state that planning documents prepared while applying Design Thinking are characterized by a greater complexity and the solutions they feature are paired better.

To prove this statement, we can ask the following additional questions:

- What defines the complexity of a conceptual planning proposal?
- When can a planning document be considered properly adapted to initial data and the expectations of the local community?
- Does the pursuit of alternative methods of design improve the quality of an assignment or project?

## 3. Materials and Methods

Attempts at introducing elements of Design Thinking into teaching were made in the years 2019 and 2020. Modules taught to second-cycle students of the Faculty of Architecture and one module from the final phase of first-cycle studies of the Spatial Management inter-faculty course appeared to be appropriate for the implementation of Design Thinking (Table 1). As per the teaching assumptions of the Cracow University of Technology, this stage features the introduction of modules that combine numerous interdisciplinary issues and present the multi-stage path of arriving at a planning proposal. A list of modules that were subjected to the study of implementing elements of Design Thinking is presented in Table 1.

**Table 1.** List of modules that formed the basis of the study.

| Item No. | Module Name | Cycle | Semester | Type of Module | Number of Hours |
|----------|-------------|-------|----------|----------------|-----------------|
| 1 | Spatial Planning | II | 2 | Design studio | 105 |
| 2 | Regional Planning | II | 2 | Laboratories | 15 |
| | | | | Lectures | 15 |
| 3 | Spatial Determinants of Water and Municipal Projects | I | 5 | Exercises | 15 |
| | | | | Lectures | 15 |

At the Cracow University of Technology, Spatial Planning [34] and Regional Planning [35] modules are taught in the form of an integrated module, during which students attempt to prepare an assignment subject of their choosing on the scale of the city and the region. The process is divided into stages which are comprised of a broad range of analyses, initial conceptual proposals and a final project—in the case of Regional Planning, due to the different specificity of its subject matter, these are proposals that, upon implementation, could allow a region to function in a more sustainable manner. In the case of the Spatial Determinants of Water and Municipal Projects module [36], students prepare design guidelines for waterfront areas and municipal infrastructure that is necessary for the functioning of urbanized areas. Practical classes are taught based on individual critiques—this allows the adaptation of the teacher's recommendations to individually chosen assignment subjects prepared by a group of students. In this respect, this method of teaching can be described as a conventional model based on the master–student relationship.

In the past year, the results achieved by groups whose teaching process included elements of Design Thinking were compared with the results achieved by a group taught without the application of this method. This comparison was prepared for the three modules under analysis, whose findings are presented in this paper.

Another assumption of the modification introduced to the teaching method was the stimulation of student creativity. Design Thinking emphasizes not only group work, but also brainstorming. It often leads to discovering unexpected alternatives that would be difficult to propose during conventional thinking about the essence of a problem. We often encounter excessive attachment to a single conceptual proposal, which is often the first to be presented. However, it often does not feature the characteristics of an analysis of the extant situation or its determinants in relation to elements like the natural environment or cultural values. Here it should be noted that it is typically linked with the need to devote a significant amount of time to preparing the aforementioned proposal using computer software tools. They offer excellent precision, but achieving it comes at the cost of significant time that must be spent on preparing a drawing. The Design Thinking method prefers simplicity and effectiveness, which are also based on the use of traditional and simple tools.

The initial overview of the term assignment was enriched by presenting Design Thinking along with a short presentation of examples of its operation and application. To introduce Design Thinking elements into the method of teaching spatial planning, the primary stages of preparing the assignment project were interspersed with short activating tasks. During such tasks, the group was to, for instance, present a large number of proposals over a short period of time. Their subject matter was adapted to each phase of the assignment so that it would be possible to make the fullest use of the process of enhanced empathy. Without effectively understanding initial data and expectations, it would be impossible to present effective solutions.

Design Thinking is based on an analysis of initial materials—the determinants of the extant state have a significant impact on identifying potential opportunities and goals assumed for the project.

In the case of the Spatial Planning and Regional Planning modules, the modules are taught to groups no larger than four persons, with one teacher being assigned no more than fifteen students. In the case of the Spatial Determinants of Water and Municipal Projects module, students work in groups of no more than four persons. One tutor is assigned to no more than 20 students.

To define the utility of introducing elements of Design Thinking, it was necessary to compare the number of solutions proposed by students during each year at each stage of preparing the term assignment. The comparison covers the years 2017 and 2018, in the case of which the term project was prepared without using Design Thinking, with the year 2019/2020, which featured elements of Design Thinking. In the final phase, the results of the group taught in accordance with a curriculum that implemented elements of Design Thinking were compared with those of the group following the standard curriculum (without elements of Design Thinking). The stages of the study for modules taught as a part of integrated design were as follows:

- An initial test (listed on charts as "test 1").
- A development directions sheet, prepared as an initial conceptual proposal.
- Tests 2 and 3 (as subsequent stages intended to verify the correctness of assumptions)—in reference to Spatial Planning due to the lower number of tests taken, the number was limited to two.
- A final project that integrated the most favorable design proposals.

Due to the different specificity of the module—particularly in reference to a more local scope of activities planned as a part of the Spatial Determinants of Water and Municipal Projects module, the verification featured three stages:

- An initial test (listed on charts as "test 1");
- An initial conceptual proposal;
- A final project.

During each stage, the focus was placed on defining the number of new, previously absent elements that can significantly affect the functioning of the project subject. The set of proposals did not account for modifications such as the following:

- Increasing development density in urbanized areas.
- Adding new areas of development.
- Supplementing the existing circulatory layout for vehicular traffic.

The solutions listed above are typically introduced to design projects in a manner that can be described as mechanical, which is why they should not be qualified as solutions that considerably enhance the functionality of space. They typically do not display an appropriate level of creativity.

The problems accounted for during grading included the following:

- Taking advantage of historical determinants—often tied with the reactivation of activities that had been taking place in the past but disappeared in recent years—including the activation of the manufacture of products unique to a given region.
- The establishment of new, legible compositional and circulatory linkages.
- The creation of interesting public spaces connected into well-defined layouts.
- The planning of continuous elements of greenery that would provide suitable conditions for isolating pedestrian and bicycle traffic from vehicular traffic. Their existence not only improves the balance of biologically active areas but allows us to count them among zones that enable active recreation.
- The implementation of new economic activity based on measures applied in the vicinity of the city that is the subject of the assignment—in the case of local activities, the adjacent area is formed by the closest fragments of urban tissue.

The amount of solutions proposed by students was recorded each time at every stage under analysis. Verification was performed via periodical reviews, which performed the role of milestones that are intended to close off certain stages of a project and enable going forward to new tasks. In this case, these are project sheets focusing on different types of subject matter. The verification of outcomes was also performed during tests, which had the students work at the university studio. The tests involved specific tasks that were to be completed in a set amount of time—these tasks could not be continued at home (Figures 5 and 6).

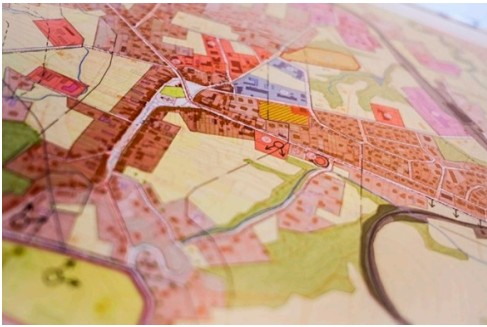

**Figure 5.** Fragment of an assignment project for the Spatial Planning project (authors: Michalina Rudol and Katarzyna Turek).

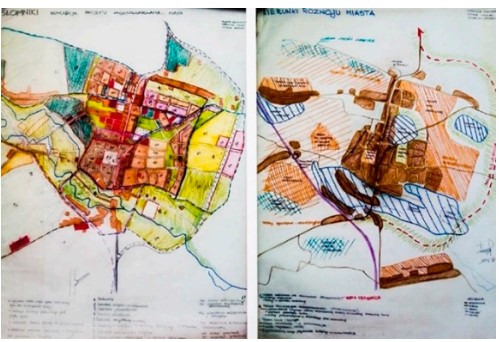

**Figure 6.** Fragment of an assignment project for the Spatial Planning project (authors: Jolanta Grzybowska, Karolina Sikora and Lucyna Smalec).

To pass the module, the students had to present a complete set of sheets and receive positive test grades. At the Cracow University of Technology, the lowest grade is 2.0, while grades between 3.0 and 5.0 allow you to complete the course. A very good grade (5.0) is the highest grade that can be obtained by a student at the Cracow University of Technology. For the final grading, the final design had to feature the following:

- Solutions to fundamental spatial problems.
- Correct circulatory solutions and accessibility to newly designed areas.
- Spatial structures and planning provisions adapted to existing determinants.
- Proposals of the placement and balancing of the quality of services that can allow the local community to function.
- Limit the negative impact of barriers and factors that negatively affect the reception and functioning of the space of the city.
- Justification for an assessment of the city's development degree, assuming that development does not have to take place, but this must be backed by arguments.

Additional criteria required to receive a good grade are as follows:

- Demonstration of the ability to solve most spatial problems.
- The use of the key assets of the natural and cultural environment so as to improve the attractiveness of urbanized space.
- Formulation of activities that positively affect the creation of new jobs and the activation of various trade groups.
- Display of the ability to activate craftsmanship and the production of goods that used to be produced in the past.

To receive a very good grade, it was necessary to include the following in the project:

- Solutions that incorporate the key assets of the natural and cultural environment into new structures thus creating favorable conditions for the development of additional activities that benefit the functioning of the city.
- Detailed guidelines in the form of a planning document draft—the guidelines should define land use and development in the area it covers in a relatively detailed manner.
- Proposal of creating a network of green and public spaces that would allow the separation of pedestrian and bicycle traffic from vehicular traffic.
- Proposals of introducing activities (that could be of a supralocal significance) that could significantly contribute to the creation of new jobs and transforming the city under design into a center whose existence will be key to the region.

A comprehensive approach to design allows the combination of various elements and results in a coherent conceptual proposal that combines numerous interdisciplinary solutions. Only in this manner is it possible to achieve the effect of synergy that can be observed in cases of good practice. The ability to solve complex spatial problems is necessary for the development of cities. Many of them still face the challenges posed by the remains of industrial buildings which are currently located in city centers and which often define the attractiveness of a given urban center.

## 4. Results

The results obtained for each year under analysis are presented in the charts below (Figures 7–9).

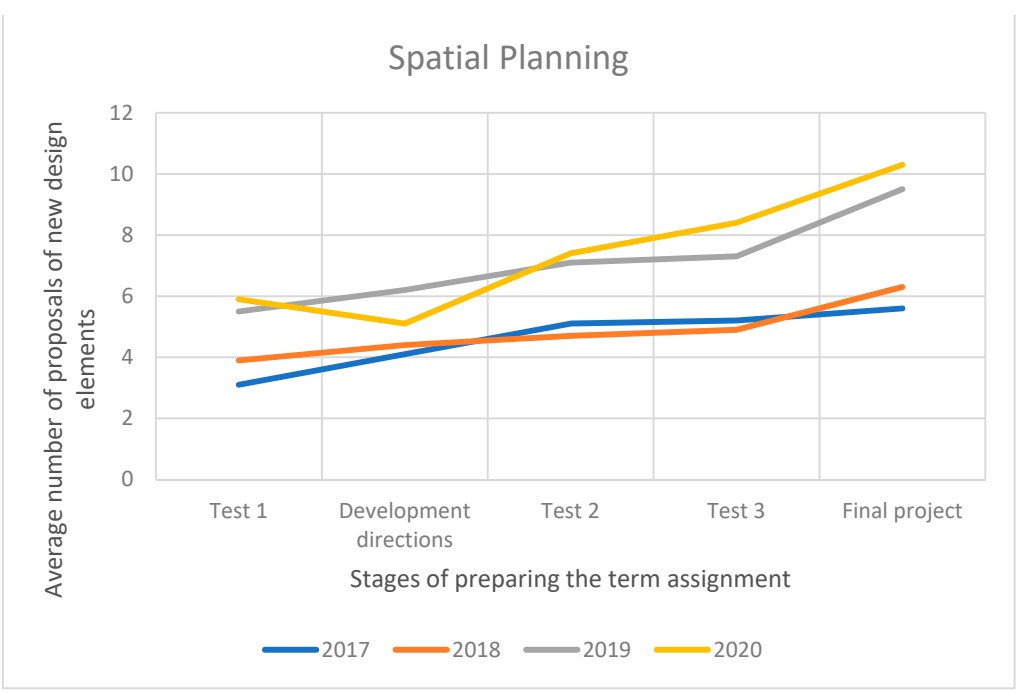

**Figure 7.** A graph showing the results obtained in the course of Spatial Planning.

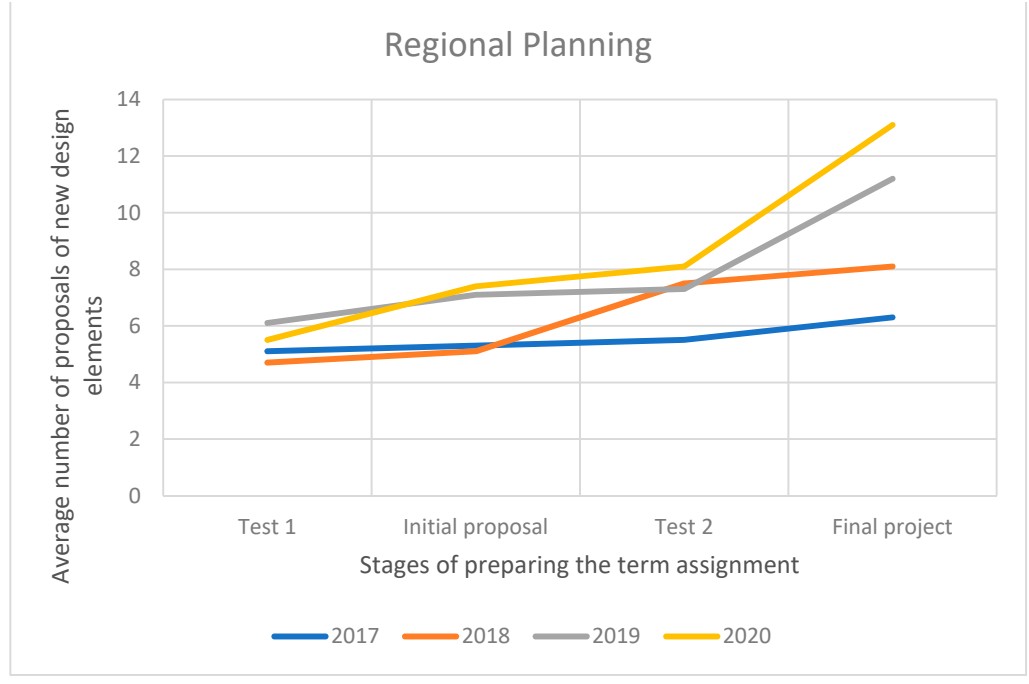

**Figure 8.** A graph showing the results obtained in the course of Regional Planning.

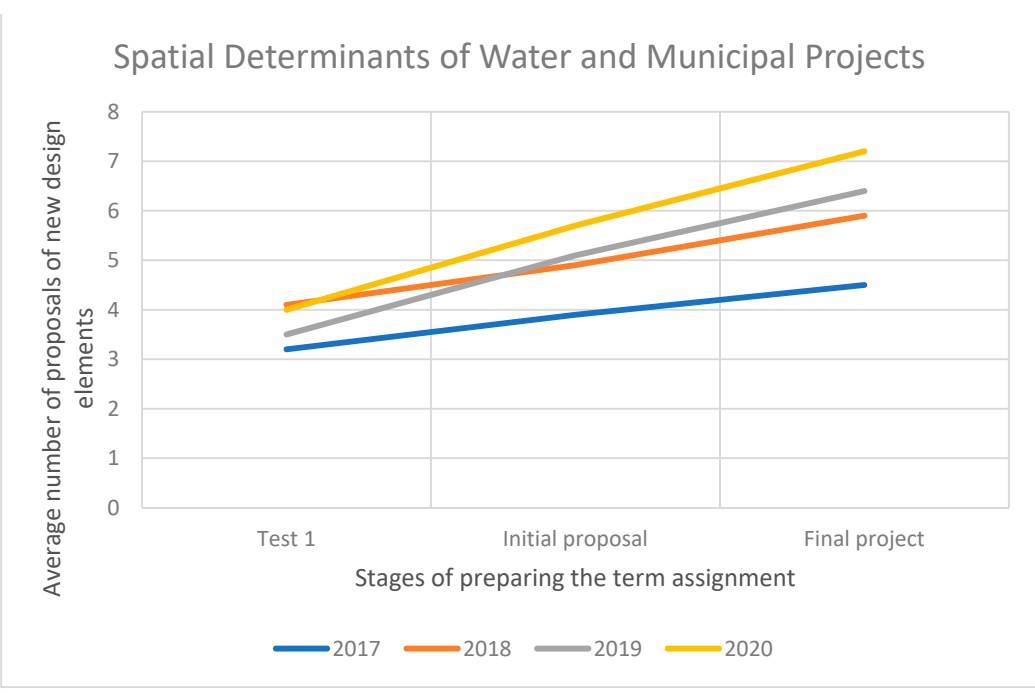

**Figure 9.** A graph showing the results obtained in the course of Spatial Determinants of Water and Municipal Projects.

The charts presented above demonstrate an overall increase of the level of complexity of assignment projects prepared by students during the period under analysis associated with the subject of this paper. The averaged results for each module are presented in the tables below (Tables 2–4).

**Table 2.** Increase in the number of highly effective design solutions as a part of the Spatial Planning module.

| Spatial Planning | | | | |
|---|---|---|---|---|
| **Test 1** | **Development Directions** | **Test 2** | **Test 3** | **Final Project** |
| 63% | 33% | 48% | 55% | 66% |

**Table 3.** Increase in the number of highly effective design solutions as a part of the Regional Planning module.

| Regional Planning | | | |
|---|---|---|---|
| **Test 1** | **Initial Proposal** | **Test 2** | **Final Project** |
| 18% | 39% | 18% | 69% |

**Table 4.** Increase in the number of highly effective design solutions as a part of the Spatial Determinants of Water and Municipal Projects module.

| Spatial Determinants of Water and Municipal Projects | | |
|---|---|---|
| **Test** | **Initial Proposal** | **Final Project** |
| 3% | 23% | 31% |

As per the assumption discussed in a previous section, a comparison of the results achieved by groups taught using a process incorporating elements of Design Thinking and the results of a group taught without such elements, in 2020, can be seen on the chart below (Figure 10).

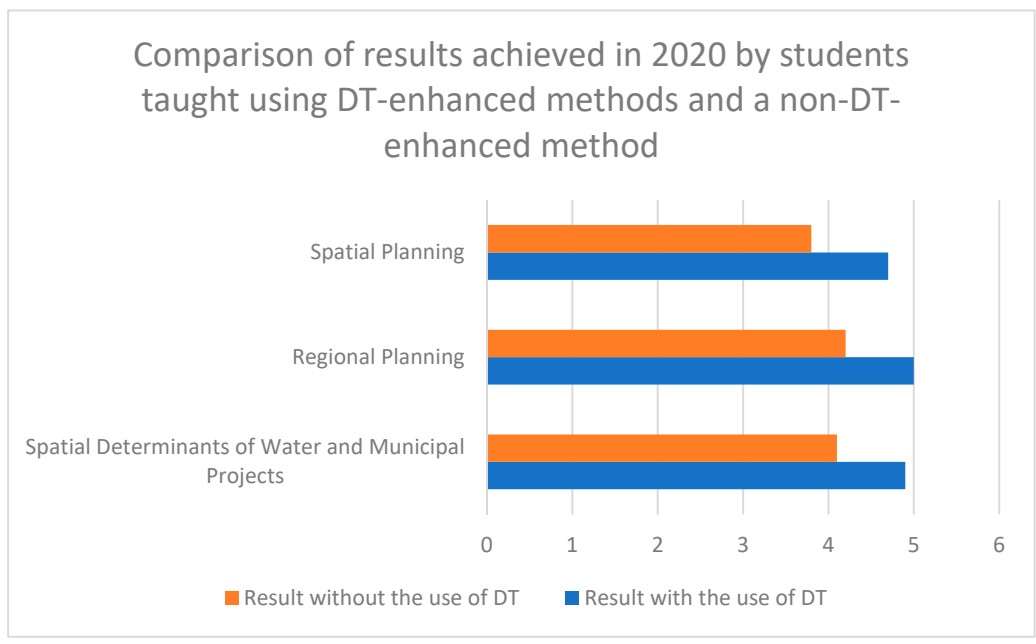

**Figure 10.** Summary of the results obtained in the subjects selected for the study.

## 5. Discussion

Every time we must remember that geometry was the reason and premise of the creation of historic ideal city projects, but their implementation in most cases turned out to be a utopia. Recording the spatial issues of a city constituting a very complex structure on the surface of paper, tracing paper or on a computer monitor is an extremely difficult conceptually task. Even the simplest graphic analysis can be modified with additional graphic symbols to supplement the record with additional but necessary information.

Currently, attempts are being made to introduce computer modeling [37] and three-dimensional printing [38] to urban design. Tools such as applications are not yet as efficient as in the case of designing architectural objects. The methods of 3D printing and terrain modeling are also not used to an appropriate degree in the process of academic education. Perhaps one of the reasons for this situation is the fact that the Local Spatial Development Plans in Poland still do not take into account such possibilities. The use of digital methods does not mean that we do not have to think about the dependencies necessary to be included in the processes related to spatial planning. Modern solutions are primarily aimed at easier visualization of the obtained results. The use of traditional design techniques allows, however, to maintain a greater dose of creativity because the student does not focus on the ways of using the tool, but on solving the given problem.

The results confirm the feasibility of introducing elements of Design Thinking into the process of teaching subject matter associated with spatial planning. It can be assumed that, apart from two exceptions, the introduction of Design Thinking into design modules resulted in an improvement in assignment project quality. The results obtained are similar depending on the teaching method used. It can be assumed that the statistical deviation remains at a similar level. The chart presented above demonstrates the dependence that, regardless of the module, a significant and very similar improvement of results achieved by students was obtained. Similar results were observed in groups of foreign students who carried out the described subjects under international exchange programs. Due to the different size of such groups, they were not graphically presented in the charts. However, the results obtained are very similar.

## 6. Conclusions

Based on the study's findings, it can be concluded that introducing elements of Design Thinking positively affects improvement in the creativity demonstrated by students. The application of Design Thinking positively contributes to improving the quality of student projects with a focus on spatial-planning-related problems. These proposals are also characterized by a more comprehensive approach by combining different elements with spatial structures and integrating economic activities in a more complex manner than in the case of projects prepared on the basis of the classical approach of formulating and developing a single conceptual proposal. Students were observed to focus more on analyzing extant determinants and the opportunities they provide, as well on formulating the proposals that make up the final project.

The effects obtained with the use of Design Thinking indicate that the number of highly effective solutions used in term projects rose by up to several dozen percent. These results are confirmed by the comparison performed during the final year under analysis, during which the results of the group working the application of Design Thinking were compared with those of a group that did not use it. The average value of student grades from both groups differed in the case of each of the three modules by almost a full grade, which is a significant difference when a five-point grading scale is used.

A certain pattern concerning modules taught using the mode of integrated design could be observed—the improvements in effectiveness were the most clearly observable in this case. This improvement was particularly pronounced in reference to grades given to projects in their final phase, but the results demonstrated throughout the semester were also very good. In the case of the Spatial Determinants of Water and Municipal Projects module, the results were not as spectacular, but they can be considered promising. In this case, the increase in projects that improved the quality of urban space rose by a third when compared to years during which classes were taught without the application of Design Thinking.

The findings confirm the hypothesis that planning documents drafted using Design Thinking are characterized by a greater complexity, and the solutions they feature are better suited to their intended use. Students who worked with Design Thinking prepared conceptual proposals that better aligned with the subject matter of sustainable development. These solutions can form a better answer (compared to projects prepared so far without the use of the Design Thinking method) to the threats presented in the latest academic publications on urban planning [24,25]. The research presented in the article is an extension of the search based on a design method that was originally created for other purposes.

Improving the quality of assignment projects and better performance in terms of optimizing a diverse array of solutions enables the enhancement of the quality of projects prepared by students, both as a part of their curriculum and during their later professional careers. Research can be used to improve the quality of teaching subjects related to spatial planning, as well as other design subjects. Their psychological aspect also illustrates the way a designer works whose mind is stimulated in various ways. These results can also be useful for analyzing ways to improve the creativity of participants of other types of courses—also outside universities.

**Author Contributions:** Conceptualization, R.B. and M.Ł.; methodology R.B. and M.Ł.; formal analysis, R.B. and M.Ł.; investigation, R.B. and M.Ł.; resources, R.B. and M.Ł.; data curation, R.B. and M.Ł.; writing—original draft preparation R.B. and M.Ł.; writing—review and editing, R.B. and M.Ł.; visualization, M.Ł.; supervision R.B. All authors have read and agreed to the published version of the manuscript.

**Funding:** This research received no external funding.

**Institutional Review Board Statement:** Not applicable.

**Informed Consent Statement:** Not applicable.

**Conflicts of Interest:** The authors declare no conflict of interest.

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
