# Peer review of "Teaching Spatial Planning Using Elements of Design Thinking as an Example of Heuristic in Urban Planning"

_sustainability, doi:10.3390/su13084225_

Round 1
Reviewer 1 Report
The manuscript addresses the topic of teaching spatial and urban planning based on the established and still innovative method of design thinking. The teaching perspective leads to a manuscript which could enrich the ongoing debates in the topic addressed. However, before considering this manuscript for publication in this international impact journal, the authors should thoroughly address the following points in a revised manuscript version.
- The Abstract is too long, and it does not end with the particular research gap / question addressed in this paper. Could you please shorten it and highlight the relevance of the teaching perspective on this topic toward the end of the abstract?
- The introduction is very extensive. In terms of the structure, could you divide it into two sections, i.e. an introduction ending up with the research problem / gap, and a background section where you present the state-of-the art literature of design thinking applied to spatial and urban planning, and its approaches in academic teaching?
- The discussion and concluding chapter hardly include any references to other related studies. In the fourth paragraph of your (final) chapter 5 (Conclusions), you state: “These solutions can form a better answer to the threats presented in the latest academic publications on urban planning [24] [25].” (ll. 543-548). This the only passage in the sections 4 (Discussion) and 5 (Conclusions) where you refer to other studies you introduced before. Could you please be more precise in terms of a) why your solutions provide a better answer than Leifer, Meinel and Rogema, and b) in how far your results extend, underline, contradict other related state-of-the-art literature?
- In your examples, you refer to 2D sketches used in planning approaches. However, there are ongoing debates which combine design thinking with modern 3D media, such as 3D prints. Moreover, there is a vivid ongoing debate on the construction of immersive virtual environments (incl. multisensory design). It would be worth mentioning in the introductory section or background section that there are many media discussed in ongoing debates of urban planning and closely related disciplines. These two references are two suggestions to show the diversity of media (from 2D multimedia maps, to 3D prints and immersive virtual reality applications).
Greenhalgh, S. (2016). The effects of 3D printing in design thinking and design education. In: Journal of Engineering, Design and Technology, 14 (4), pp. 752-769. https://doi.org/10.1108/JEDT-02-2014-0005
Edler, D., Kühne, O., Keil, J., Dickmann, F. (2019). Audiovisual Cartography: Established and New Multimedia Approaches to Represent Soundscapes. In: KN – Journal of Cartography and Geographic Information, 69 (1): 5-17. https://doi.org/10.1007/s42489-019-00004-4
- A minor aspect: You should remove your academic titles from your names. I have hardly seen any titles before names in international research manuscripts.
Author Response
Dear Reviewer.
The abstract has been shortened to the required length not exceeding 200 words according to the guidelines for the authors. We have also changed the content to better match the description of the article and the research it contains. (line number 9).
Thank you for paying attention to the length of the introduction - it should indeed be shorter. The introduction was divided into a short part with an introduction and the second one containing research background, which, according to the authors, significantly influences the understanding of the problem. We hope that this action had a positive effect on organizing the structure of the document. (line number 123 (after accepting the changes, it is a line number 48).
The implementation of the Design Thinking method we applied allowed us to increase the level of creativity of students. We wanted to check whether the methods often proposed in other areas of the search for form can actually translate into the study of spatial planning at the faculty of architecture. The Design Thinking method was established as a method of searching for a form for creating everyday objects. As mentioned above - our research is an extension of the research presented in the aforementioned publications (especially with reference to the literature provided in the earlier part of the article). (lines number 584-588 (after accepting the changes, it is a lines number 554-558).
Thank you for providing interesting sources. We got acquainted with them and included them in the discussion in the article. Currently, however, such solutions are not commonly used in Poland in relation to spatial planning. Most municipalities and cities try to supplement the area coverage with local spatial development plans, which often refer only to small areas. Currently, local spatial development plans in Poland are not adequately reflected in digital data. The above suggestion of including digital modelling methods may be an idea for us to conduct further research in terms of introducing them to the curriculum at the university - not only as part of classes on the use of computer programs but also in terms of their application in spatial planning. (lines number 530-540 (after accepting the changes, it is a lines number 500-510).
As suggested by the reviewer, the academic titles were removed. (line number 6).
The entire text has been linguistically checked again, so if there are any mistakes, please indicate them.
Best regards
Authors
Reviewer 2 Report
The abstract in the online system and the pdf article are not the same.
Please correct the abstract, in the pdf version it is too long and detailed. Follow the requirements: the abstract contains a summary of the entire paper and can be up to 200 words long, gives background and motivation to the paper, a brief description of the methods, the principle results, then conclusions or interpretations.
If you use the abbreviation, mention it in the first use, e.g. Design Thinking (DT) and then, use it consistently further in the article (applies also for abstract).
Please separate the description of methodological steps and criteria from the results (descriptions methodological steps and criteria move from the Results to Methods).
It is not clear, what are highly effective design solutions, those which received the best grade A?
Author Response
Dear Reviewer,
The abstract has been shortened to the required length not exceeding 200 words according to the guidelines for the authors. We have also changed the content to better match the description of the article and the research it contains. (line number 9).
We resigned from using the abbreviation in order to unify the content throughout the article and to keep one designation consistently.
The obtained results were separated from the description of methods and criteria. We agree with the correctness of this remark - the change in the division allowed for a better organization of the content. (line number 488 (after accepting the changes, it is line number 458).
The best grade that can be obtained by a student at the Cracow University of Technology is a very good grade (5.0) - that is why the term "very good" was used, which is related to the 5-point grading scale at the University (subject to grades from 2 to 5). Added information that grade 5 is the highest possible. (line number 444 (after accepting the changes, it is line number 413).
The entire text has been linguistically checked again, so if there are any mistakes, please indicate them.
Best regards
Authors
Reviewer 3 Report
In the publication, the authors point to the possibilities of enriching the ways of educating architecture students. The proposal to use elements of Design Thinking in teaching spatial planning is another extension of the designer's workshop. The problem remains the legislative aspect of spatial planning, which causes difficulties in cooperation between different professions.
Author Response
Dear Reviewer
The Design Thinking method was originally created for a purpose other than the preparation of spatial planning projects. As shown in the literature and our research, this method can also be useful for creating urban projects. This method is not widely used by architects and planners - regardless of the size of the project team. We believe that introducing it to the curriculum positively influences the improvement of the quality of projects - which has been confirmed by our research. Legal conditions do not apply here as they are not the subject of research or part of the title. In addition, an interesting project that takes into account the needs of the local community may find greater support when agreeing on building conditions or a local plan (both in talks with the investor and during public consultations). The article does not focus on cooperation between different professions, but the Design Thinking method is very often used in teams combining different professions to work out the most optimal solution.
Best regards
Authors
Round 2
Reviewer 1 Report
The authors provided a revised version of the manuscript. The structure was changed which has a high and improving impact on the readability. In addition, authors have paid attention to the suggestions made in the first review round. Changes are explained in a response letter. The letter could have been given a better structure along the points of the reviewer. However, the general thoughts are transported.
Reviewer 2 Report
The authors clarified points that were not clear and significantly improved the article.